# ENHANCING TOOL CALLING IN LLMS WITH THE INTERNATIONAL TOOL CALLING DATASET

## ABSTRACT

Tool calling allows large language models (LLMs) to interact with external systems like APIs, enabling applications in customer support, data analysis, and dynamic content generation. Despite recent advances, challenges persist due to limited datasets with simulated or inaccessible APIs and insufficient geographic diversity. To address this, we present the International Tool Calling (ITC) dataset, designed for real-world, international tool calling scenarios. ITC includes 3,571 real APIs and 17,540 tool calling tasks across 20 categories and 40 countries. The dataset was constructed through a four-stage pipeline: API collection and construction, query generation, query scoring and filtering, and question–answer pair generation. Experiments reveal substantial performance gaps between open- and closed-source LLMs, while fine-tuning on the full multilingual ITC dataset significantly improves generalization, cross-lingual reasoning, and out-of-domain tool usage. ITC provides a valuable benchmark for advancing LLM robustness and performance in complex, multi-tool, and international scenarios. Dataset: https://anonymous.4open.science/r/International-Tool-Calling-ITC-dataset-FAF4/.

## 1  INTRODUCTION

Tool calling empowers large language models (LLMs) to interact with external systems—such as databases, APIs, and software tools—extending their capabilities beyond text generation (Schick et al., 2023). By invoking tools, LLMs can access real-time data, perform complex computations, and execute actions beyond their training data (Nakano et al., 2021). This functionality is essential for tasks such as automated customer support, data analysis, and dynamic content generation, where external resource integration enhances both performance and utility. As surveyed in (Mialon et al., 2023), tool calling enables more sophisticated, context-aware interactions, making LLMs valuable across diverse domains.

Recent advances have led to the development of several datasets and benchmarks to improve tool-use capabilities in LLMs. Notable examples include API-BLEND (Basu et al., 2024), APIGen (Liu et al., 2024c), and ToolACE (Liu et al., 2024b), which focus on API-based function calling across a variety of use cases. Others, such as Gorilla (Patil et al., 2023) and ToolLLM (Qin et al., 2023), address real-world tool invocation, reducing hallucinations and improving accuracy. More complex datasets like Seal-Tools (Wu et al., 2024), PLUTO (Huang et al., 2024), and SciToolBench (Ma et al., 2024) explore multi-step reasoning and domain-specific tool use. Collectively, these benchmarks have advanced the development of LLMs capable of interacting with external tools effectively.

However, key challenges remain. Many existing datasets rely on simulated APIs, lacking the complexity and variability of real-world tool usage (e.g., Seal-Tools (Wu et al., 2024)). Others use proprietary or inaccessible APIs, as in ToolLLM (Qin et al., 2023), limiting reproducibility and real-world deployment. Accessibility is further hindered by datasets that are not publicly available. Moreover, most existing benchmarks are US-centric, making them unsuitable for region-specific tasks in a global context. For example, while APIs like Yahoo_Weather can retrieve data for major cities like Shenzhen, they often fail to provide detailed, district-level information (e.g., Nanshan), underscoring the need for broader geographic coverage and finer granularity.

To overcome existing limitations in tool calling research, we present the **International Tool Calling (ITC)** dataset, specifically designed to support real-world, globally distributed tool calling scenar-

ios. The final dataset comprises 3,571 real-world APIs and 17,540 tool calling tasks—15,790 for training and 1,750 for testing—covering 20 categories across 40 countries. It includes 64.2% global APIs—such as machine translation and international weather services—and region-specific APIs from major regions like the United States and China, along with 38 additional countries, ensuring broad geographic and functional diversity. By encompassing a wide range of single- and multi-tool tasks, ITC captures realistic challenges in tool selection, parameter specification, and cross-cultural usage, making it a comprehensive resource for evaluating and improving the performance and generalization of tool-augmented language models.

We benchmarked 16 open-source and 8 closed-source LLMs on the ITC test set, revealing substantial performance gaps across multiple metrics and highlighting common challenges in tool usage, such as handling nonexistent tools, missing parameters, and incorrect parameter generation. Fine-tuning on the full multilingual ITC dataset yields significant performance gains, particularly on non-English queries, by enhancing reasoning consistency and cross-lingual generalization, while also improving out-of-domain generation and boosting tool selection and invocation precision on external benchmarks, demonstrating ITC's effectiveness in enhancing generalization and robustness in complex, real-world scenarios.

## 2 RELATED WORK

Existing benchmarks for enhancing LLM tool-invocation cover a variety of tasks, including API-based interactions, multi-step reasoning, and robustness evaluation. Datasets such as API-BLEND (Basu et al., 2024), APIGen (Liu et al., 2024c), and ToolACE (Liu et al., 2024b) provide diverse APIs for training and evaluation, with APIGen and ToolACE emphasizing executable APIs, while API-BLEND focuses on semantic parsing and slot-filling. Gorilla (Patil et al., 2023) and ToolLLM (Qin et al., 2023) improve LLM performance on real-world API interactions, and Seal-Tools (Wu et al., 2024), PLUTO (Huang et al., 2024), and SciToolBench (Ma et al., 2024) introduce more complex tool-use scenarios. Other datasets, including RoTBench (Ye et al., 2024c), StableToolBench (Guo et al., 2024), ToolEyes (Ye et al., 2024a), and ToolSword (Ye et al., 2024b), evaluate robustness, safety, and cognitive capabilities, while multi-modal frameworks like MLLM-Tool (Wang et al., 2024) extend interactions to images, text, and audio.

Table 1: Summary of existing tool calling datasets.

| Dataset | # Tools | Source | Accessibility | # Tasks | Executability | Primary language |
|---|---|---|---|---|---|---|
| API-BLEND | 199 | Simulated | × | 189,040 | × | English |
| APIGen | 3,673 | Real | × | 60,000 | √ | English |
| Gorilla | 1,645 | Real | √ | 16,450 | √ | English |
| Seal-Tools | 4,076 | Simulated | √ | 14,076 | × | English |
| ToolACE | 26,507 | Simulated | × | 11,300 | × | English |
| ToolBench | 16,464 | Real | √ | 126,486 | √ | English |
| RoTBench | 568 | ToolEyes | × | 105 | √ | English |
| MLLM-Tool | 932 | Real | × | 11,642 | √ | English |
| PLUTO | 2,032 | Simulated | × | 5,824 | × | English |
| SciToolBench | 2,446 | Simulated | × | 856 | × | English |
| GeoLLM-QA | 117 | Real | × | 1,000 | × | English |
| INJECAGENT | 17 | Simulated | √ | 1,054 | √ | English |
| StableToolBench | 16,464 | ToolBench | √ | 126,486 | √ | English |
| ToolEyes | 568 | Simulated | √ | 382 | √ | English |
| ToolSword | 100 | Simulated | × | 440 | √ | English |
| Hammer | - | APIGen | × | 67,500 | × | English |
| **ours** | **3,571** | **Real** | √ | **17,540** | √ | **Multi-languages** |

Table 1 summarizes representative tool-calling datasets. Despite progress, existing benchmarks face limitations: many rely on simulated APIs that lack real-world variability, some real-API datasets use proprietary or inaccessible endpoints, over half are not publicly available, and most are US-centric, limiting global applicability. Our dataset addresses these issues by providing 3,571 real, publicly accessible APIs across multiple languages and domains.

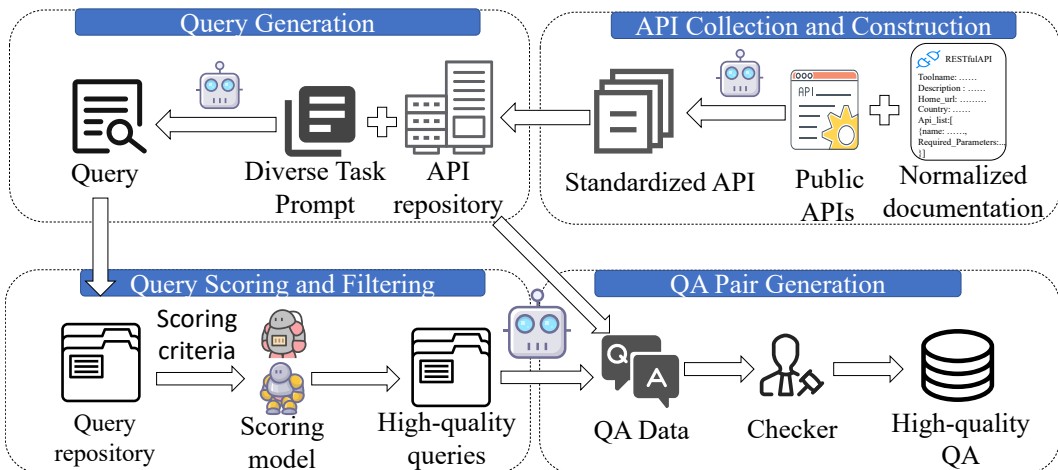

Figure 1: Dataset construction flowchart.

# 3 DATASET CURATION

With minimal human intervention, our pipeline first collects API documentation (**Stage 1**), then uses GPT-4o to generate detailed API instructions (**Stage 2**). Next, Claude-3.5-Sonnet and Gemini-1.5-Pro refine queries for clarity and executability (**Stage 3**), and finally GPT-4o generates high-quality QA pairs (**Stage 4**). This design enables easy expansion to new APIs (Figure 1).

## 3.1 API COLLECTION AND CONSTRUCTION

**API Collection**: We gathered 49,937 real REST APIs across domains such as social media, e-commerce, and weather from global sources including (RapidAPI, 2025), (Juhe, 2025), (XiaRou, 2025), and community-maintained GitHub repositories such as (Free-api, 2025) and (Public-apis, 2025). The APIs are organized into 20 functional categories to ensure broad coverage.

**API Supplementation**: To ensure reliable LLM parsing, all API documents are standardized into a uniform schema covering name, description, endpoint, method, authentication, and input/output parameters (see Figure 3, Appendix A.2). GPT-4o, guided by tailored prompts (Figure 6), is used to complete and refine incomplete or unclear specifications based on available metadata. Correctness is verified through sample executions, and APIs with irreparable issues are removed. This process produces consistent, high-quality API specifications that enhance usability for both LLMs and developers (example in Figure 16, Appendix A.10).

**API Filtering**: We applied a two-stage filtering to ensure quality and reliability. In Stage 1, automated scripts tested each API with predefined queries, removing non-responsive APIs, those returning errors (e.g., 404, 500), empty responses, or malformed/non-JSON outputs, reducing the pool from 49,937 to 5,410 (examples in Figures 4 and 5, Appendix A.2). In Stage 2, remaining APIs were assessed for stability (3–5 calls over 48 hours), update frequency (discarding APIs inactive for over 12 months), and response informativeness (task-specific queries), resulting in 3,571 high-quality APIs (7.1% of the original), stable, actively maintained, and suitable for generating tool-use instructions.

## 3.2 QUERY GENERATION

We categorize tool-calling tasks into *Single Tool Calling*, which invokes a single API, and *Multiple Tools Calling*, coordinating several APIs. The latter includes *Repeated* (same API multiple times), *Parallel* (multiple APIs simultaneously), and *Nested* (chained API calls) subtypes. Real-world scenarios often require multilingual and region-aware capabilities—for example, a Japanese tourist planning a trip to Lijiang in China may need local weather and travel information from a Chinese API, with both queries and responses in Japanese. To support such use cases and generate high-quality queries, we construct a multilingual, region-specific *seed pool* covering all task

types. Starting from the seed pool, queries are generated via an API-focused process. For each seed scenario, APIs are selected from our multilingual repository according to three principles: (1) **Geographic diversity**: include APIs from countries or regions that have fewer available APIs, so the dataset is not dominated by a few regions; (2) **Functional variety**: include APIs that perform similar or complementary tasks, allowing repeated, parallel, or chained calls in a scenario; (3) **Disambiguation challenge**: include APIs with similar names or outputs to test whether the model can choose the correct one in context. For each API (or API set), GPT-4o generates three user queries conditioned on 1–3 task-specific seed examples (see Appendix A.4 and A.5). This ensures queries remain faithful to the seed scenarios, match API functionality, and cover diverse languages and regions.

## 3.3 QUERY SCORING AND FILTERING

In the previous step, we obtained 44,198 generated queries, many of which suffered from unclear requirements, low relevance, non-standard language, or poor adherence to cultural context. Our query selection involved two steps: **Query Scoring** and **Query Filtering**. In the scoring step, we evaluated each query across five dimensions—Relevance, Practicality, Linguistic Applicability, Clarity, and Specificity (Appendix A.6.1)—using two independent LLMs, Claude-3.5-Sonnet and Gemini-1.5-Pro, with scores from 1 (lowest) to 5 (highest). During **Query Filtering**, only queries scoring above 4 from both models were retained, removing 25,830 queries (58.4%) and leaving 18,368 candidates. To ensure quality, five multilingual annotators manually reviewed all remaining queries using the same five criteria, with each query checked by at least two annotators and disagreements resolved through discussion. After this verification, 17,540 queries remained, representing a further 4.5% removal. This two-stage process—automatic scoring followed by exhaustive human review—ensures that the final queries are highly relevant, clear, practical, specific, and linguistically appropriate, making the dataset reliable for downstream tasks.

## 3.4 QUESTION-AND-ANSWER PAIR GENERATION

Based on the 17,540 curated queries generated in the previous step and their associated tools, we used GPT-4o to generate complete answers. For each query, we first identified its task type—**Single**, **Repeated**, **Parallel**, or **Nested Tool Calling**—and applied task-specific prompt templates (Appendix A.7). Single, Repeated, and Parallel tasks were generated in a single turn, while Nested tasks, which involve multi-step reasoning, were generated via a controlled, step-by-step process to ensure logical consistency. Each answer was paired with its query to form a QA instance, yielding 17,540 QA pairs. To ensure quality, we employed an LLM-based *Checker* system using Claude-3.5-Sonnet and Gemini-1.5-Pro, evaluating *consistency* between reasoning and API calls, *solution validity*, and *linguistic quality*. Low-complexity issues were corrected directly by humans, while high-complexity tasks were reviewed jointly by the Checker and human annotators. In total, 1,214 QA pairs (6.9%) were modified, producing the final high-quality dataset of 17,540 QA pairs (see Appendix A.8 for Checker prompts and details).

## 4 DATA STATISTICS

Our International Tool Calling (ITC) dataset comprises 3,571 APIs and a total of 17,540 question-and-answer pairs, including 15,790 for training and 1,750 for testing. In the following sections, we detail the composition of the dataset from two perspectives: APIs and Tasks.

The APIs in the ITC Dataset are systematically organized into 20 functional categories, derived from their official descriptions, core functionalities, and typical usage scenarios, as illustrated in Figure 2a. The dataset is dominated by a few key categories. The largest is **Finance**, which accounts for 14.25% of the total APIs. This is followed by **Data** at 12.9%, **Communication** at 9.75%, and **Entertainment** at 8.18%. In contrast, the smallest categories show a minimal presence. These include **Travel** with only 0.22%, and both **Math** and **Sports** each representing 0.84% of the APIs. This wide disparity highlights a significant concentration of APIs in finance, data, communication, and entertainment, while other areas are sparsely represented.

Global APIs—such as those supporting machine translation and international weather forecasting—constitute the majority of our dataset, with 2,291 samples (64.2%), primarily from providers

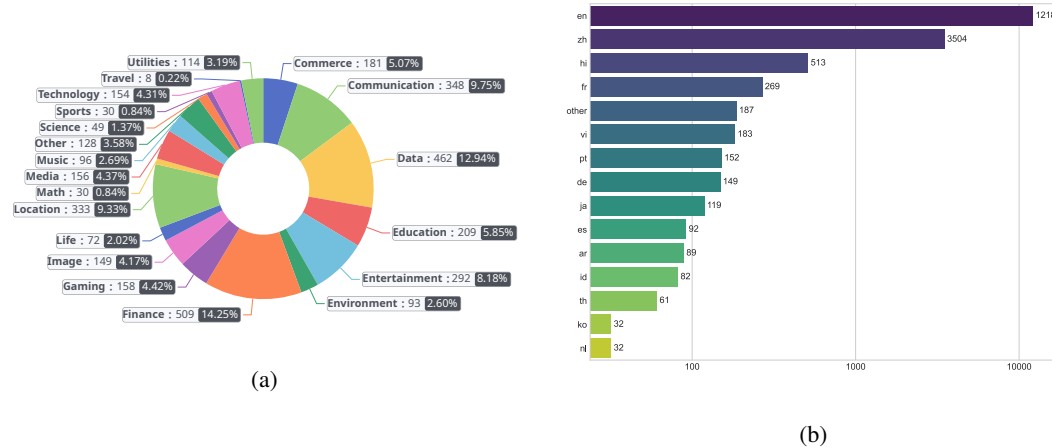

(a)

(b)

Figure 2: (a) API category distribution in the ITC Dataset; (b) Language distribution of all tasks. For clarity, languages with fewer than 30 occurrences are aggregated into the "other" category. The x-axis is presented on a logarithmic scale.

based in the United States. In addition, country-specific APIs offer localized services such as regional weather and news, with strong representation from major regions including China and the United States, which together contribute 61.79% of the samples in this category. The dataset also includes APIs from 38 additional countries, contributing to regional diversity and capturing a broad spectrum of localized functionalities worldwide. A detailed distribution is shown in Figure 7 in Appendix A.2.5.

Our dataset consists of 17,540 tasks, including 14,295 single-tool calling tasks and 3,245 multiple-tool calling tasks. The language distribution of all tasks is shown in Figure 2b. English is the most prevalent language, accounting for 12,187 tasks (69.48%). This dominance is primarily due to the large proportion of global APIs originating from the United States and the widespread use of English as a lingua franca in API documentation. In addition to English, the dataset contains a rich diversity of 28 other languages. A complete breakdown of all 29 languages and their respective counts is provided in Appendix A.9.

## 5 EXPERIMENTS AND RESULTS

### 5.1 IMPLEMENTATION DETAILS

Our experiments involved both open-source and closed-source large language models (LLMs). The open-source models, which are publicly available for research and development, include general-purpose models such as **Qwen2.5** (Yang et al., 2024) and **DeepSeek-V3** (Liu et al., 2024a), as well as models specifically designed for tool calling, such as **Hammer2** (Lin et al., 2024) and **Watt-tool-8B**. In contrast, the closed-source group comprises state-of-the-art proprietary models, such as **GPT-4o**, **Claude-3.5-Sonnet**, and **o3-mini**. For evaluation on our dataset, open-source models were tested using their default configurations. For fine-tuning, we adopted the LoRA framework (Hu et al., 2021), training each model for 3 epochs with a batch size of 1 per device and 8 gradient accumulation steps. The learning rate was set to 1.0e-4, and we employed a cosine learning rate scheduler with a warmup ratio of 0.1. This setup ensures stable convergence while adapting the models to the tool calling tasks in our dataset.

### 5.2 EVALUATION METRICS

To comprehensively evaluate model performance, we adopt four evaluation metrics. The first three are based on the Seal-Tools framework (Wu et al., 2024): **(1) Tool Selection (P/R/F1):** Measures the model's ability to accurately identify the appropriate tool(s) from a set of candidates. Performance is evaluated using precision, recall, and F1-score, reflecting tool localization accuracy; **(2) Tool In-**

**vocation (P/R/F1):** Assesses the model's ability to generate correct and complete tool invocation parameters. We compute precision, recall, and F1 based on triple-level matching of the tool name, parameter key, and parameter value; **(3) Format Matching Accuracy (FM):** Evaluates whether the model's output conforms to the expected JSON schema. This is a critical requirement for ensuring compatibility with downstream execution environments. While these metrics capture key aspects of tool calling, they overlook an essential requirement in multilingual, real-world applications: maintaining linguistic consistency throughout the interaction. To address this gap, we introduce a new metric: **(4) Language Matching Accuracy (LM):** Quantifies the proportion of cases in which the model's internal reasoning (i.e., the *thought* field) is expressed in the same language as the user's input query. We use the `langid` library for language identification. Detailed formulations and implementation details for all four metrics are provided in Appendix A.3.

Table 2: Zero-shot evaluation on ITC test data (%), best results in bold.

| Type | Model Name | LM | FM | Tool Selection | | | Tool Invocation | | |
|------|-----------|-----|-----|------|------|------|------|------|------|
| | | | | **P** | **R** | **F1** | **P** | **R** | **F1** |
| Open-source | Qwen2.5-7B-Instruct | 90.51 | 96.65 | 54.08 | 53.06 | 53.18 | 42.76 | 43.37 | 42.71 |
| | Qwen2.5-Coder-7B | 94.93 | 98.38 | 69.76 | 66.01 | 67.23 | 54.17 | 54.11 | 53.75 |
| | Qwen2.5-3B-Instruct | 87.40 | 93.00 | 49.34 | 45.84 | 47.52 | 40.90 | 41.77 | 41.33 |
| | Qwen2.5-Coder-3B | 84.26 | 89.25 | 48.92 | 49.01 | 48.76 | 38.49 | 38.83 | 38.43 |
| | Watt-tool-8B | 74.48 | 5.53 | **88.90** | **88.03** | **88.30** | **76.33** | **73.46** | **74.31** |
| | ToolACE-8B | 81.31 | 4.56 | 70.30 | 69.82 | 69.93 | 59.39 | 56.22 | 57.17 |
| | Hammer2.1-7b | 86.82 | 20.71 | 64.64 | 64.68 | 64.44 | 33.14 | 32.68 | 32.75 |
| | Hammer2.0-7b | 78.21 | 95.42 | 61.22 | 57.48 | 58.68 | 45.00 | 45.25 | 44.85 |
| | Functionary-v3.1 | 76.75 | 54.15 | 40.63 | 37.15 | 38.30 | 35.25 | 35.64 | 35.02 |
| | Yi-1.5-9B-Chat-16K | 82.37 | 91.9 | 45.28 | 45.71 | 45.32 | 35.67 | 35.66 | 35.33 |
| | GLM-4-9B-Chat | 76.00 | 97.55 | 43.45 | 41.44 | 42.09 | 32.77 | 32.85 | 32.57 |
| | Phi-4 | **96.73** | 96.29 | 80.90 | 82.68 | 81.49 | 70.15 | 70.25 | 69.84 |
| | Qwen2.5-Coder-32B | 91.05 | 99.14 | 84.82 | 81.44 | 82.54 | 71.13 | 71.04 | 70.69 |
| | Qwen2.5-72B-Instruct | 89.47 | 98.16 | 52.78 | 51.44 | 51.83 | 43.11 | 43.35 | 42.89 |
| | DeepSeek-V3 | 86.09 | 99.89 | 83.10 | 83.73 | 83.28 | 75.94 | 75.77 | 75.49 |
| | DeepSeek-R1 | 77.05 | **100** | 86.89 | 85.25 | 85.79 | 73.47 | 73.15 | 72.79 |
| Closed-source | o1-mini | 95.89 | 93.68 | 64.41 | 66.61 | 64.72 | 60.58 | 62.53 | 61.06 |
| | o3-mini | 86.19 | 71.37 | 61.06 | 61.13 | 60.93 | 54.01 | 53.56 | 53.54 |
| | GPT4o-mini | 96.24 | **99.83** | 76.47 | 75.21 | 75.55 | 71.69 | 70.38 | 70.71 |
| | GPT4o | **97.95** | 99.83 | **88.95** | **89.48** | **89.01** | **82.18** | **81.57** | **81.57** |
| | GLM-Zero | 88.37 | 98.45 | 51.24 | 50.31 | 50.51 | 42.64 | 43.64 | 42.78 |
| | Gemini-2.0-Flash | 95.04 | 99.77 | 77.25 | 77.76 | 77.32 | 69.08 | 68.14 | 68.18 |
| | Gemini-2.0-Pro | 96.17 | 94.13 | 84.57 | 83.50 | 83.86 | 73.22 | 71.65 | 71.95 |
| | Claude-3.5-Sonnet | 94.75 | 97.06 | 82.08 | 81.00 | 81.19 | 72.05 | 72.29 | 71.77 |

## 5.3 ZERO-SHOT EVALUATION OF TOOL CALLING CAPABILITIES

We evaluate the zero-shot performance of large language models (LLMs) on the ITC test set to assess their intrinsic tool calling capabilities without task-specific fine-tuning.

**Overall performance:** Table 2 presents results across four key metrics: Language Matching (LM), Format Matching (FM), Tool Selection, and Tool Invocation. Overall, closed-source models consistently outperform open-source models. `GPT-4o` achieves the highest LM (97.95%) and FM (99.83%) scores, followed closely by `Claude-3.5-Sonnet` and `Gemini-2.0-Pro`. Among open-source models, `Deepseek-V3` and `Qwen2.5-Coder-32B` perform well, achieving FM above 99% and LM above 86%. In contrast, models such as `Watt-tool-8B` achieve strong task-level performance but suffer from low LM (74.48%) and FM (5.53%), indicating weaknesses in multilingual handling and structural adherence. Lower-performing models like `Functionary-v3.1` and `Hammer2.1-7B` struggle across all dimensions, producing outputs that are often malformed or inconsistent with user language.

**Linguistic and structural accuracy:** For **LM**, most closed-source models exceed 95%, with `GPT-4o` at 97.95%, while open models like `Qwen2.5-7B-Instruct` (90.51%) and `Phi-4` (96.73%) also perform well. For **FM**, `Deepseek-R1` reaches 100%, most closed-source models exceed 95%, and over two-thirds of open models meet the requirements. Models like

`Watt-tool-8B` and `ToolACE-8B` have low FM because they generate only tools and parameters without the multi-step reasoning traces required by ITC, causing misformatted or incomplete JSON outputs.

**Functional competence in tool calling:** Closed-source models demonstrate strong overall capabilities in both tool selection and invocation. `GPT-4o` achieves the highest performance across both tasks, with F1 scores of 89.01% for Tool Selection and 81.57% for Tool Invocation. Other proprietary models such as `Gemini-2.0-Pro` and `Claude-3.5-Sonnet` also show robust competence, each attaining F1 scores above 80% in both metrics—indicating reliable tool identification and parameter synthesis. Among open-source models, `Watt-tool-8B` stands out with a Tool Selection F1 of 88.30% and an Invocation F1 of 74.31%, rivaling the top-performing closed models. `Deepseek-V3` and `Qwen2.5-Coder-32B` also deliver competitive performance, demonstrating balanced strength across both tool recognition and structured invocation generation. In contrast, models such as `Hammer2.1-7B` and `Functionary-v3.1` exhibit limited functional competence. Their Tool Invocation F1 scores, 32.75% and 35.02% respectively, reveal substantial weaknesses in generating correct and executable tool calls. The performance gap between Tool Selection and Tool Invocation serves as an important diagnostic signal. High-performing models like `GPT-4o` exhibit consistent accuracy across both stages, suggesting well-integrated reasoning and output synthesis. Conversely, models with large disparities—such as `Hammer2.1-7B`—often succeed at selecting relevant tools but fail to construct valid invocation arguments. This discrepancy reflects challenges in multi-step planning, schema adherence, and structured output formatting. For real-world deployment, such inconsistencies can compromise system reliability, as both decision accuracy and execution correctness are critical.

Table 3: Error analysis results across different LLMs (%), best results in bold.

| Type | Model Name | Tool Selection | | | Tool Invocation | | |
|------|-----------|-------------|---------|-------|-----------|---------|-------|
| | | Hallucinating | Missing | Extra | Incorrect | Missing | Extra |
| Open-source | Qwen2.5-7B-Instruct | 21.57 | 73.23 | 5.20 | 51.53 | 19.73 | 28.74 |
| | Qwen2.5-Coder-7B | 4.25 | 86.65 | 9.10 | 51.01 | 20.59 | 28.39 |
| | Qwen2.5-3B-Instruct | 8.74 | 75.79 | 15.47 | 42.14 | **16.42** | 41.45 |
| | Qwen2.5-Coder-3B | 38.48 | **51.51** | 10.00 | 37.36 | 23.02 | 39.62 |
| | Watt-tool-8B | 25.51 | 67.74 | 6.74 | 45.54 | 40.61 | 13.85 |
| | ToolACE-8B | 4.12 | 88.75 | 7.13 | 42.03 | 48.63 | **9.34** |
| | Hammer2.1-7b | 0.70 | 91.56 | 7.74 | **17.18** | 64.26 | 18.56 |
| | Hammer2.0-7b | 2.35 | 89.16 | 8.49 | 57.80 | 23.92 | 18.28 |
| | Functionary-v3.1 | 20.92 | 76.97 | **2.11** | 37.70 | 28.80 | 33.51 |
| | Yi-1.5-9B-Chat-16K | 37.35 | 55.78 | 6.86 | 38.90 | 18.66 | 42.44 |
| | GLM-4-9B-Chat | 0.98 | 93.90 | 5.12 | 27.88 | 54.78 | 17.34 |
| | Phi-4 | 11.29 | 69.18 | 19.53 | 44.75 | 29.11 | 26.14 |
| | Qwen2.5-Coder-32B | 8.09 | 66.31 | 25.61 | 46.43 | 27.80 | 25.78 |
| | Qwen2.5-72B-Instruct | 43.64 | 51.95 | 4.41 | 46.65 | 22.79 | 30.56 |
| | DeepSeek-V3 | **0.43** | 80.21 | 19.36 | 57.65 | 24.38 | 17.97 |
| | DeepSeek-R1 | 8.33 | 83.33 | 8.33 | 41.67 | 29.17 | 29.17 |
| Closed-source | o1-mini | 35.5 | 61.68 | **2.82** | 54.5 | 28.5 | 17.0 |
| | o3-mini | 34.8 | 62.33 | 2.87 | 53.89 | 29.64 | 16.47 |
| | GPT4o-mini | 19.71 | 76.26 | 4.03 | 48.76 | 39.94 | **11.29** |
| | GPT4o | 47.16 | **49.72** | 3.12 | 53.67 | 21.22 | 25.10 |
| | GLM-Zero | 37.63 | 58.09 | 4.28 | 48.89 | **13.27** | 37.83 |
| | Gemini-2.0-Flash | 30.93 | 62.13 | 6.95 | 50.42 | 29.50 | 20.08 |
| | Gemini-2.0-Pro | **0** | 85.64 | 14.36 | **47.31** | 35.13 | 17.56 |
| | Claude-3.5-Sonnet | 22.11 | 68.81 | 9.08 | 54.67 | 21.21 | 24.12 |

**Error Analysis:** Table 3 breaks down tool calling errors into *Tool Selection* (hallucinating non-existent tools, missing required tools, selecting extra tools) and *Tool Invocation* (generating incorrect, missing, or extra parameters). In the selection phase, missing required tools is the most frequent error: `Gemini-2.0-Pro` hallucinates 0% but misses 85.64%, reflecting a highly conservative strategy, whereas `GPT-4o` maintains a more balanced profile with 47.16% hallucinations and 49.72% missing tools. Open-source models such as `Qwen2.5-Coder-3B` show higher hallucinations (38.48%) but lower missing rates, indicating aggressive yet imprecise selection. In invocation, incorrect or missing parameters are common; for example, `Hammer2.1-7B` has a low incorrect

parameter rate (17.18%) but misses 64.26% of arguments, occasionally misassigning values and violating API schemas. Overgeneration of extra tools or parameters is less frequent but observed in models like `Qwen2.5-Coder-32B` and `Phi-4`. Models such as `Gemini-2.0-Pro` and `ToolACE-8B` exhibit balanced error distributions, reflecting stronger robustness and better structured reasoning, while models like `Watt-tool-8B` and `Functionary-v3.1` tend to over- or under-specify outputs, highlighting weaknesses in schema adherence and multi-step planning. Overall, these results demonstrate that both tool omission and parameter errors remain critical challenges for reliable real-world tool calling.

## 5.4 FINE-TUNED EVALUATION OF TOOL CALLING CAPABILITIES

In this experiment, we fine-tuned four Qwen2.5 models—`Qwen2.5-7B-Instruct`, `Qwen2.5-Coder-7B`, `Qwen2.5-3B-Instruct`, and `Qwen2.5-Coder-3B`—as well as two DeepSeek models—`DeepSeek-Coder-7B-Instruct-v1.5` and `DeepSeek-Coder-1.3B-Instruct`—on our ITC training dataset to assess improvements in tool calling capabilities for open-source LLMs.

Table 4: Evaluation of fine-tuned models on the ITC test set (%), with improvements over the original models in brackets. $Q_1$: Qwen2.5-7B-Instruct, $Q_2$: Qwen2.5-Coder-7B-Instruct, $Q_3$: Qwen2.5-3B-Instruct, $Q_4$: Qwen2.5-Coder-3B-Instruct, $D_1$: DeepSeek-Coder-7B-Instruct-v1.5, $D_2$: DeepSeek-Coder-1.3B-Instruct.

| Model | LM | FM | Tool Selection | | | Tool Invocation | | |
|---|---|---|---|---|---|---|---|---|
| | | | P | R | F1 | P | R | F1 |
| $Q_1$ | 96.9(+6.4) | 99.8(+3.1) | **97.7**(+43.6) | **98.1**(+45.0) | **97.8**(+44.6) | **90.6**(+47.9) | **90.6**(+47.2) | **90.3**(+47.6) |
| $Q_2$ | **97.4**(+2.5) | 99.6(+1.3) | 97.7(+27.9) | 98.0(+32.0) | 97.7(+30.5) | 90.6(+36.4) | 90.4(+36.3) | 90.2(+36.5) |
| $Q_3$ | 97.3(+9.9) | 99.5(+6.5) | 97.4(+48.0) | 97.9(+52.1) | 97.5(**+50.0**) | 89.8(+48.9) | 89.5(+47.7) | 89.4(+48.0) |
| $Q_4$ | 97.3(**+13.0**) | **99.8**(+10.5) | 97.6(+48.7) | 97.9(+48.9) | 97.6(+48.9) | 90.3(**+51.8**) | 90.3(**+51.4**) | 90.0(**+51.5**) |
| $D_1$ | 77.4(+3.6) | 78.7(+32.5) | 76.5(+51.2) | 76.9(+51.0) | 76.5(+51.2) | 68.4(+48.6) | 68.2(+48.2) | 68.0(+48.4) |
| $D_2$ | 77.9(+7.9) | 79.3(**+59.9**) | 56.4(**+53.1**) | 56.9(**+53.6**) | 56.4(+53.1) | 46.4(+44.2) | 46.1(+43.9) | 45.9(+43.7) |

**ITC test set results:** Table 4 shows substantial gains in both tool selection and tool invocation after fine-tuning across all evaluated models. For Qwen, the fine-tuned 3B variants achieve performance comparable to the larger 7B variants. For example, `Qwen2.5-7B-Instruct` improved tool selection recall by 45.0% and tool invocation precision by 47.9%, while `Qwen2.5-Coder-3B` recorded the largest boost in tool invocation F1 at 51.5%. These results demonstrate the effectiveness of our training dataset in enhancing tool calling performance across model scales. For DeepSeek, fine-tuning also brings notable gains, with the 7B model outperforming the 1.3B variant across all metrics, achieving up to 51.0% improvement in tool selection F1 and 48.0% in tool invocation F1. However, their limited multilingual support and weaker instruction-following leave them trailing the Qwen models on most metrics.

Table 5: Evaluation of fine-tuned models on three external benchmarks (%), showing tool selection precision (TS) and tool invocation precision (TI); values in parentheses indicate improvements, with best results and largest gains in bold. $Q_1$: Qwen2.5-7B-Instruct, $Q_2$: Qwen2.5-Coder-7B-Instruct, $Q_3$: Qwen2.5-3B-Instruct, $Q_4$: Qwen2.5-Coder-3B-Instruct.

| Model | Nexus Raven | | Seal-Tools | | Tool-Alpaca | |
|---|---|---|---|---|---|---|
| | TS | TI | TS | TI | TS | TI |
| $Q_1$ | 90.6(**+25.8**) | 60.0(+10.2) | 89.9(**+24.0**) | 76.2(+17.8) | **77.1**(**+18.1**) | 50.0(+9.9) |
| $Q_2$ | **91.0**(+20.4) | **68.0**(**+17.8**) | 89.6(+22.2) | **78.0**(+18.9) | 77.3(+14.9) | **50.9**(+8.2) |
| $Q_3$ | 81.0(+6.5) | 57.1(+2.8) | **90.3**(+23.3) | 76.8(**+20.0**) | 75.0(+8.9) | 47.5(+8.6) |
| $Q_4$ | 84.2(+2.1) | 64.2(+4.9) | 89.3(+8.5) | 76.2(+8.0) | 73.1(+4.5) | 48.7(+7.1) |

**Out-of-domain generalization:** To evaluate robustness beyond the training distribution, we tested the fine-tuned Qwen2.5 models on several external benchmarks (Table 5). All models exhibit marked improvements, with tool selection precision increasing up to 25.8% and tool invocation precision im-

proving by up to 18.1%. This indicates that fine-tuning not only strengthens in-domain capabilities but also enhances generalization to unseen tools and tasks.

## 5.5 Ablation Study on Language Impact

Table 6: Effect of multilingual vs. English-only training on Qwen2.5 models for non-English ITC queries (%), with improvements in brackets; best results and largest gains in bold. 'ALL' = full dataset, 'EN' = English subset. $Q_1$: Qwen2.5-7B-Instruct, $Q_2$: Qwen2.5-Coder-7B-Instruct, $Q_3$: Qwen2.5-3B-Instruct, $Q_4$: Qwen2.5-Coder-3B-Instruct.

| Type | Model | LM | FM | Tool Selection | | | Tool Invocation | | |
|---|---|---|---|---|---|---|---|---|---|
| | | | | **P** | **R** | **F1** | **P** | **R** | **F1** |
| ALL | $Q_1$ | 96.3(+5.6) | 99.3(+4.2) | 91.6(+36.7) | **98.6(+42.0)** | **94.9**(+39.7) | 87.8(+45.9) | 86.4(+44.1) | 87.1(+45.5) |
| | $Q_2$ | **96.5**(+7.0) | **98.9**(+1.5) | **93.3**(+21.8) | 93.6(+19.6) | 93.4(+20.7) | **88.4**(+33.2) | **89.2**(+34.8) | **88.8**(+34.6) |
| | $Q_3$ | 91.6(**+11.2**) | 95.9(**+13.6**) | 87.0(+39.6) | 89.8(+40.3) | 88.4(**+39.9**) | 76.2(+44.1) | 74.2(+41.3) | 75.2(+42.7) |
| | $Q_4$ | 94.2(+9.3) | **98.9**(+4.4) | 87.4(**+40.2**) | 86.9(+39.3) | 87.2(+39.9) | 80.2(**+46.9**) | 80.4(**+45.7**) | 80.3(**+46.7**) |
| EN | $Q_1$ | 91.3(+0.6) | 97.1(+2.0) | 79.6(+24.7) | 79.1(+22.6) | 79.3(+24.1) | 70.8(+29.0) | 71.2(+28.9) | 71.0(+29.4) |
| | $Q_2$ | **92.6**(+2.1) | **98.3**(+1.2) | **88.2**(+15.7) | **88.3**(+14.4) | **88.3**(+15.5) | **79.5**(+24.3) | **79.6**(+25.2) | **79.6**(+25.3) |
| | $Q_3$ | 83.6(**+3.3**) | 86.5(**+4.2**) | 78.0(+30.5) | 77.7(+28.2) | 77.8(+29.4) | 69.7(**+37.6**) | 70.0(**+37.0**) | 69.8(**+37.3**) |
| | $Q_4$ | 85.7(+0.8) | 96.5(+1.9) | 78.9(**+31.6**) | 79.5(**+31.9**) | 79.2(**+31.9**) | 69.9(+36.5) | 70.0(+35.2) | 69.9(+36.3) |

To investigate the effect of non-English data, we fine-tuned Qwen2.5 models on either the full multilingual ITC dataset (`Type = ALL`) or the English-only subset (`Type = EN`) and evaluated them on non-English test queries. As Table 6 shows, models trained on the full multilingual dataset consistently outperform English-only models across all metrics. For instance, `Qwen2.5-7B-Instruct` achieves a +42.0% gain in tool selection recall versus +22.6% for English-only training, while `Qwen2.5-Coder-7B` improves tool invocation F1 by +34.6%, exceeding English-only performance by over 9%. Multilingual training also enhances LM and FM scores, reflecting better alignment with query language and improved structured output. These results indicate that incorporating diverse language data not only boosts cross-lingual generalization but also strengthens reasoning consistency and parameter formatting, which are crucial for robust real-world tool calling.

To assess the impact of non-English languages on model performance, we conducted an ablation study by fine-tuning Qwen2.5 models either on the full multilingual ITC training set ('Type = ALL') or exclusively on the English subset ('Type = EN'), then evaluating on the non-English test data. As shown in Table 6, models trained on the full dataset achieve substantially higher gains on non-English queries. For example, the Qwen2.5-7B-Instruct model fine-tuned on all languages improved tool selection recall by 42.0%, which is 19.4% higher than the gain achieved by the same model trained only on English. Similarly, tool invocation F1 for Qwen2.5-Coder-7B increased by 34.6% with full multilingual training, outperforming the English-only training gain by 9.3%. These results demonstrate that limiting training to English significantly restricts performance gains on non-English queries, highlighting the importance of incorporating diverse language data to strengthen the multilingual generalization capabilities of LLMs.

## 6 Conclusion

In this paper, we introduce the International Tool Calling (ITC) dataset, a diverse and globally representative resource aimed at advancing large language models' capabilities in multi-tool and international API scenarios. Covering a broad range of API categories, ITC addresses challenges such as long-tail API coverage and complex multi-tool interactions. Our experiments show that fine-tuning on ITC leads to substantial performance gains, including improvements on out-of-domain tasks, demonstrating its effectiveness in enhancing LLMs' interaction with international APIs. Nevertheless, ITC has limitations: some regions remain underrepresented, it currently covers only REST APIs, and reliance on free APIs may affect stability. Future work should expand geographic coverage, include additional tool types, and develop more challenging benchmarks to further improve the robustness, inclusivity, and general applicability of tool-calling systems.

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

# A    APPENDIX

## A.1    THE USE OF LARGE LANGUAGE MODELS (LLMS)

In accordance with the ICLR 2026 policy on transparency, we disclose that Large Language Models (LLMs) were a core, integral component of our methodology for creating the International Tool Calling (ITC) dataset. LLMs were fundamental to the semi-automated, four-stage pipeline developed for its construction, as described in Section 3. We detail their precise roles at each stage below.

### A.1.1    ROLE IN THE ITC DATASET CURATION PIPELINE

**Stage 1: API Standardization and Supplementation.**    As detailed in Section 3.1, we used **GPT-4o** to process the 49,937 initially collected raw API documents. Guided by tailored prompts (see Figure 6), the model parsed unstructured or incomplete information and reformatted it into our standardized JSON schema. This involved inferring and completing missing descriptions, standardizing parameter types, and ensuring the final API specifications were consistent and machine-readable for subsequent stages.

**Stage 2: Query Generation.**    In the second stage (Section 3.2), **GPT-4o** was responsible for generating the initial, large-scale pool of user queries. For each selected API or set of APIs, the model was prompted with task-specific seed examples to generate diverse queries that covered single-tool, multi-tool (repeated, parallel, nested), multilingual, and region-specific scenarios. This automated generation was essential for achieving the scale and diversity of the final dataset.

**Stage 3: Query Scoring and Filtering.**    To programmatically filter the 44,198 generated queries, we employed two independent LLMs, **Claude-3.5-Sonnet** and **Gemini-1.5-Pro**, as automated evaluators (Section 3.3). Each query was scored by both models across five predefined dimensions (Relevance, Practicality, Clarity, etc.). This LLM-based scoring system enabled us to efficiently identify and discard over half of the initial queries, focusing human annotation efforts on higher-quality candidates.

**Stage 4: Question-Answer Pair Generation and Quality Checking.**    In the final stage (Section 3.4), **GPT-4o** was used to generate the complete QA pairs. For each curated query, it produced the "thought" process and the structured tool call(s) in the required JSON format. Subsequently, an LLM-based **Checker** system, powered by **Claude-3.5-Sonnet** and **Gemini-1.5-Pro**, was implemented to perform quality control. The Checker automatically validated the logical consistency, solution validity, and linguistic quality of the generated pairs, flagging instances that required manual review and correction by human annotators.

While LLMs were instrumental in scaling our data creation process, we emphasize that every stage involving LLM generation was designed with human oversight. All automated outputs were systematically reviewed, filtered, and refined by the authors. The authors take full responsibility for the final content of the dataset and this manuscript, including its integrity and accuracy.

## A.2 API Processing

### A.2.1 API Format

```
API Format

{
    "tool_name":"tool name",
    "tool_description":"tool description",
    "home_url":"home url",
    "country":"Countries involved in the tool",
    "api_list":[
        {
            "name":"api name",
            "url":"api url",
            "description":"api function description",
            "method":"POST/GET",
            "required_parameters":[
                {
                    "name":"parameter name",
                    "type":"parameter type",
                    "description":"parameter description",
                    "default":"Default value, empty if none"
                }
            ],
            "optional_parameters":[
                {
                    "name":"parameter name",
                    "type":"parameter type",
                    "description":"parameter description",
                    "default":"Default value, empty if none"
                }
            ],
            "statuscode":"statuscode"
        }
    ]
}
```

Figure 3: API Format.

### A.2.2 API Error Response

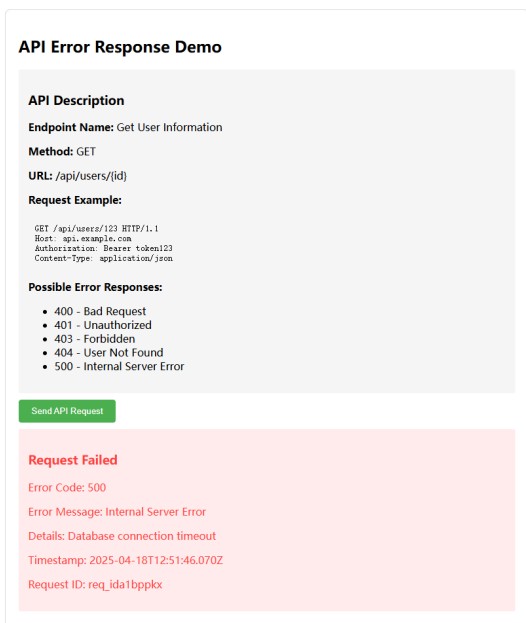

Figure 4: API Error Response Demo.

### A.2.3 API EMPTY RESPONSE

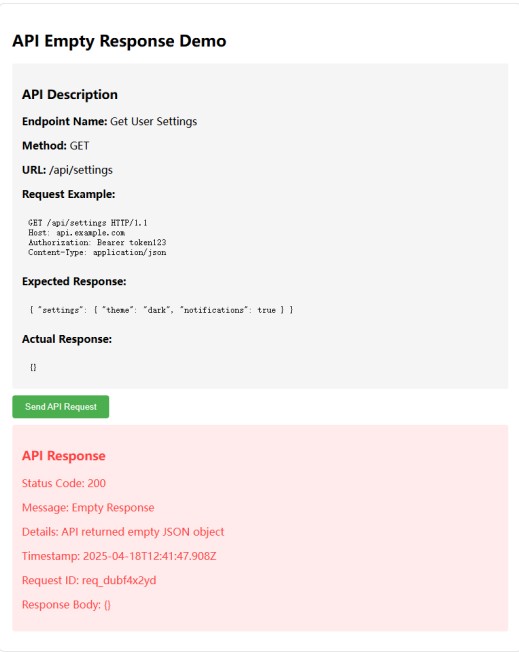

Figure 5: API Empty Response Demo.

### A.2.4 API DOCUMENTATION REWRITING

```
API Documentation Rewriting Prompt

You are a highly skilled data engineer specializing in API
analysis, able to accurately identify the functionality of
APIs and their various parameters, and excel at extracting
information from APIs.
You have the following API information:
{tool_info}

Your task:
Rewrite the API based on the provided format example. The
format example is as follows:
```json
{tool_example}
```

Note:
1. If any descriptions are missing or unclear, you need to fill
them in with simple and understandable information.
2. Parameter information must strictly follow the provided API
details, and you should not fabricate any false parameters.
3. The output should still be in JSON format with no extra
content.

your response:
```

Figure 6: API Documentation Rewriting.

### A.2.5 API COUNTRY DISTRIBUTION

Figure 7 provides a comprehensive overview of the geographical distribution of APIs in our dataset, including both global and country-specific APIs across more than 30 countries and regions.

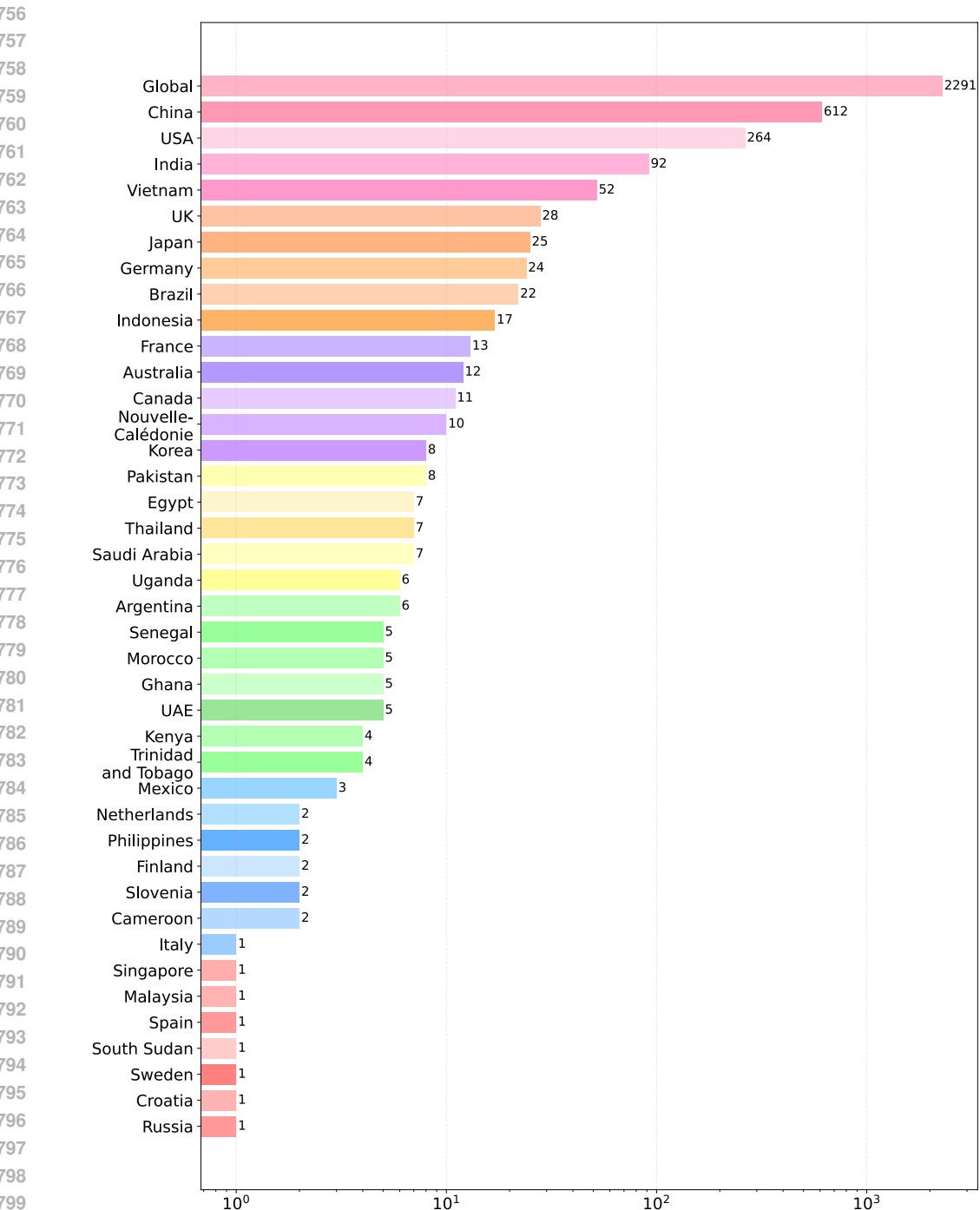

Figure 7: Distribution of tools by countries (Logarithmic Scale).

### A.3 Detailed Formulate for Evaluation Metrics

To control page layout, we use **FM** to represent *Format Matching Accuracy*, **LM** for *Language Matching Accuracy*, **Tool** for *Tool Selection*, and **TI** for *Tool Invocation*.

$$LM = \frac{amount_{correct\ language}}{amount_{all}} \quad (1)$$

$$FM = \frac{amount_{correct\ format}}{amount_{all}} \tag{2}$$

$$Tool\ P = \frac{amount_{correct\ tools}}{amount_{predict\ tools}} \tag{3}$$

$$Tool\ R = \frac{amount_{correct\ tools}}{amount_{gold\ tools}} \tag{4}$$

$$Tool\ F1 = \frac{2 \cdot Tool\ P \cdot Tool\ R}{Tool\ P + Tool\ R} \tag{5}$$

$$TI\ P = \frac{amount_{correct\ parameters}}{amount_{predict\ parameters}} \tag{6}$$

$$TI\ R = \frac{amount_{correct\ parameters}}{amount_{gold\ parameters}} \tag{7}$$

$$TI\ F1 = \frac{2 \cdot TI\ P \cdot TI\ R}{TI\ P + TI\ R} \tag{8}$$

### A.4 SINGLE TOOL CALLING TASKS QUERY GENERATION

For single tool calling tasks, we utilize a prompt-based approach to instruct the LLM to generate a query. The prompt templates used for this process are illustrated in Figures 8.

---

**Single Tool Calling Tasks Query Generation Prompt**

Please strictly follow these guidelines: 1. The instructions should be 1 to 2 sentences long. Use a mix of interrogative sentences, first-person statements, imperative sentences, and other structures that convey a request.Aim for diversity in your instructions.
2. Do not mention the API's name in your instructions.
3. Your instructions should only involve the features provided by these APIs.
4. Generate 10 diverse instructions.
5. Use specific nouns and real-world examples from various domains, such as entertainment, sports, or technology.
6. Please provide concrete details.Don't using any form of generic phrases, such as "this xxx", "the xxx","a xxx" or "a specific xxx".
7. Ensure diversity in language by combining questions with imperative statements and other structures that convey a request.
8. The instructions should be in the language of the country attribute in the provided API information.
9. The generated problem must strictly follow the API's parameter information.
10. If country is Global, please generate 10 instructions in English.

Here is the API information:
{example_list}

Please generate the question in the language of the specified country.
your response:

---

Figure 8: Query generation prompt for single tool calling tasks.

### A.5 MULTIPLE TOOLS CALLING TASKS QUERY GENERATION

For multiple tool calling tasks, we have classified them into three categories: Repeated Calls, Parallel Calls, and Nested Calls. Given that the requirements for each type of task differ, we have tailored specific prompts to generate queries for each category. The prompt templates for these tasks are illustrated in Figures 9, 10, and 11.

---

**Multiple tool Repeated Calling Tasks Query Generation**

Please strictly follow these guidelines: 1. Provide 1 to 2 sentences for each instruction, using a mix of interrogative sentences, imperative statements, and descriptive requests. Ensure the instructions are diverse in structure and tone to reflect real-world scenarios.
2. For each instruction, select only one tool (from the provided API list) and perform multiple calls to the same tool to complete different tasks.
3. Do not mention any API names directly in your instructions. Instead, focus on the functionality they provide.
4. Leverage the unique features of the selected tool. Each instruction must clearly demonstrate how the tool can be used through multiple calls to effectively solve a problem or fulfill a request. Avoid generic or vague task descriptions.
5. Use detailed and specific nouns, examples, and contextual scenarios from domains like travel, business, sports, entertainment, or technology. Avoid vague phrases such as "this information" or "a specific detail."
6. The generated instructions must strictly align with the parameter information of each API/tool. Ensure the inputs and outputs are valid for the respective tools.
7. Generate 10 diverse instructions, each showcasing a single tool being used multiple times. Each instruction can use a different tool.
8. Ensure the content of each instruction strictly aligns with the examples provided and closely follows the format of the examples below.

Here are some examples for Multi-Tool Instructions:
{example_list}

Here is the API information:
{api_data}

Here is the Output format:
{Output_format}

Please select only one tool (from the provided API list) and strictly following the Output format.

your response:

Figure 9: Multiple tool repeated calls.

---

**Multiple tool Parallel Calling Tasks Query Generation**

Please strictly follow these guidelines:
1. Provide 1 to 2 sentences for each instruction, using a mix of interrogative sentences, imperative statements, and descriptive requests. Ensure the instructions are diverse in structure and tone to reflect real-world scenarios.
2. For each instruction, select only one tool (from the provided API list) and perform multiple calls to the same tool to complete different tasks.
3. Do not mention any API names directly in your instructions. Instead, focus on the functionality they provide.
4. Leverage the unique features of the selected tool. Each instruction must clearly demonstrate how the tool can be used through multiple calls to effectively solve a problem or fulfill a request. Avoid generic or vague task descriptions.
5. Use detailed and specific nouns, examples, and contextual scenarios from domains like travel, business, sports, entertainment, or technology. Avoid vague phrases such as "this information" or "a specific detail."
6. The generated instructions must strictly align with the parameter information of each API/tool. Ensure the inputs and outputs are valid for the respective tools.
7. Generate 10 diverse instructions, each showcasing a single tool being used multiple times. Each instruction can use a different tool.
8. Ensure the content of each instruction strictly aligns with the examples provided and closely follows the format of the examples below.

Here are some examples for Multi-Tool Instructions:
{example_list}

Here is the API information:
{api_data}

Here is the Output format:
{Output_format}

Please select only one tool (from the provided API list) and strictly following the Output format.

your response:

Figure 10: Multiple tool parallel calls.

## A.6 QUERY SCORING

### A.6.1 SCORING DIMENSIONS

To comprehensively assess the quality of instructions (queries or question-and-answer pairs), we adopt the following five evaluation dimensions:

1. **Relevance**: Measures the alignment between the instruction and the task objective. High-scoring instructions accurately reflect the task requirements, while irrelevant or off-topic instructions receive lower scores.

2. **Practicality**: Assesses the feasibility and executability of the instruction in real-world scenarios. High scores indicate instructions that can be directly implemented without significant obstacles.

```
Multiple tool Nested Calling Tasks Query Generation

Please strictly follow these guidelines:
1. Provide 1 to 2 sentences for each instruction, using a mix of interrogative sentences,imperative statements, and descriptive requests. Ensure the
instructions are diverse in structure and tone to reflect real-world scenarios.
2. For each instruction, select exactly 2 or 3 tools from the provided API list to create a scenario where these tools are used in a logical sequence. Ensure
that the output of the previous tool can serve as the input for the next tool, forming a nested call.
3. Do not mention any API names directly in your instructions. Instead, focus on the functionality they provide.
4. Use detailed and specific nouns, examples, and contextual scenarios from domains like travel, business, sports, entertainment, or technology. Avoid
vague phrases such as "this information" or "a specific detail."
5. The generated instructions must strictly align with the parameter information of each API/tool. Ensure the inputs and outputs are valid for the respective
tools.
6. Generate 10 diverse instructions, each involving 2 or 3 tools working together in a logical sequence and existence of nested calls.
Here are some examples for Multi-Tool Instructions:
{example_list}

Here is the API information:
{api_data}

Here is the Output format:
{Output_format}

Please strictly following the Output format.

your response:
```

Figure 11: Multiple tool nested calls.

3. **Linguistic Applicability**: Evaluates the instruction's adherence to grammatical norms and consideration of cultural and linguistic context. High-scoring instructions are well-phrased, natural, and unambiguous.

4. **Clarity**: Judges whether the instruction is clearly articulated, logically coherent, and easy to understand. High scores indicate concise, explicit, and actionable instructions.

5. **Specificity**: Measures the level of detail and focus in the instruction. High-scoring instructions clearly define the scope of operation, reduce ambiguity, and facilitate precise tool invocation.

Each dimension is scored on a scale from 1 to 5, where 1 indicates very low quality and 5 indicates very high quality. The detailed scoring criteria are shown in Table 7

Table 7: Scoring guidelines for each evaluation dimension.

| Dimension | 1 (Very Low) | 3 (Medium) | 5 (Very High) |
|---|---|---|---|
| Relevance | Completely irrelevant | Partially relevant | Highly aligned with the task |
| Practicality | Hard to implement | Feasible but with obstacles | Directly applicable |
| Linguistic Applicability | Incorrect or awkward | Basically fluent | Standard, natural, idiomatic |
| Clarity | Vague or confusing | Understandable but ambiguous | Clear and unambiguous |
| Specificity | Overly broad or vague | Some details present | Detailed, focused, unambiguous |

### A.6.2 EXAMPLE OF SCORING

Figure 12 illustrates an example of query scoring, where, given a query and relevant API information, we used both Anthropic's Claude-3.5-sonnet model and Google's Gemini-1.5-pro model to evaluate the query's quality across five dimensions, with scores ranging from 1 to 5 for each dimension. Figure 13 shows the prompt for LLMs to evaluate the query.

### A.7 QA GENERATION

To further evaluate the model's ability to employ APIs as external tools in multilingual settings, we design a dedicated *QA Generation* prompt.Specifically, the response format requires two compo-

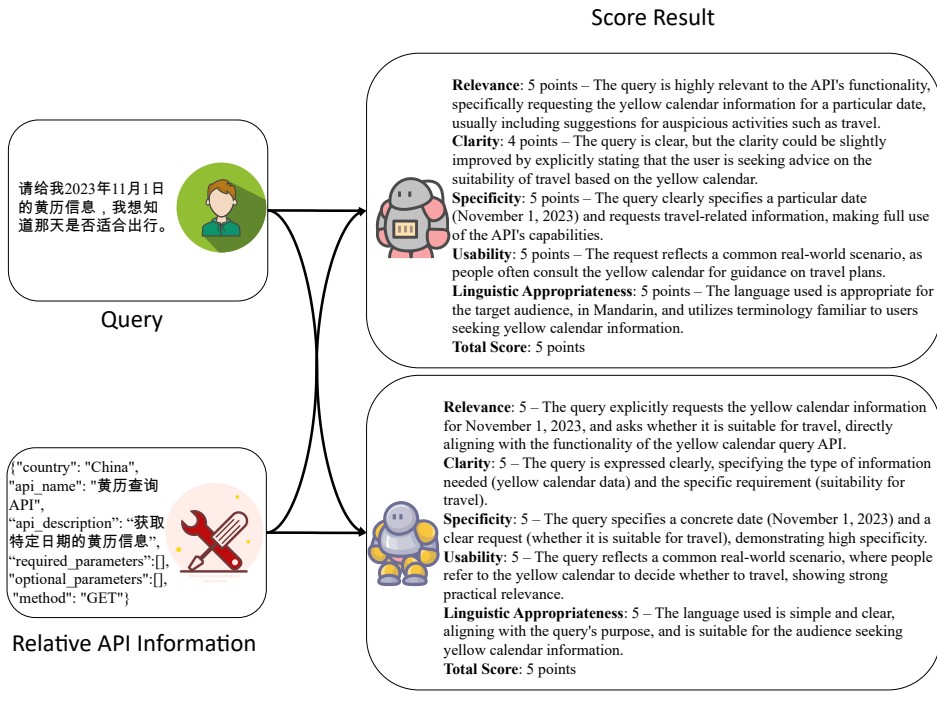

Figure 12: The query scoring process.

nents: `Thought`, which captures the intermediate reasoning steps, and `Action`, which specifies the chosen API call along with the necessary parameters. Additionally, the model is instructed to answer strictly in the language specified by the provided `country` attribute, ensuring robustness in multilingual environments. The complete prompt template is presented in Figure 14.

## A.8 CHECKER

To ensure data quality during the multi-turn QA pair generation process, we designed and introduced an LLM-based Checker module for quality filtering. This module is used to determine whether automatically generated QA pairs meet the following criteria:

- **Consistency**: Whether the question and answer are semantically aligned, and whether the answer is genuinely based on the API response.

- **Reasonability**: Whether the answer reasonably reflects the tool's output and avoids fabrication.

- **Linguistic Quality**: Whether the sentence is fluent and grammatically correct.

### A.8.1 IMPLEMENTATION DETAILS OF THE CHECKER

We used Claude-3.5-sonnet and Gemini-1.5-pro as the primary quality assessment model. The prompt is as illustrated in Figures 15. We set the temperature of the Checker to 0 to ensure stability in its judgments.

## A.9 FULL LANGUAGE DISTRIBUTION

Table 8 provides the complete list of all 29 languages present in our dataset, along with the exact number of tasks for each language.

---

**Query Scoring Prompt**

Evaluation Criteria: Use a 1-5 scale to score the following five dimensions:
1. Relevance: How well the query matches the API's functionality.
2. Clarity: Whether the query is specific enough, avoiding ambiguous terms like 'this xxx', 'the xxx', or 'a xxx', and ensuring the use of the API's features.
3. Specificity: Whether the query is specific enough to utilize the API's capabilities
4. Practicality: Whether the query reflects real-world usage scenarios
5. Language Appropriateness: Whether the query's language is suitable for target users

Scoring Standard:
1 point: Does not meet the standard
2 points: Partially meets the standard
3 points: Meets the basic standard
4 points: Meets the standard well
5 points: Fully meets the standard

Total Score Calculation:
Calculate the average of the five dimension scores, round to the nearest integer, as the final total score (1-5 points).

Evaluation Steps:
1. Carefully read the API name and the generated query.
2. Score each dimension and provide a brief explanation.
3. Calculate the total score.
4. Provide an overall evaluation and suggestions for improvement.
5. If the total score is less than 3, mark it as "Needs Improvement".

Output Format:
Scores:
1. Relevance: [Score] - [Explanation]
2. Clarity: [Score] - [Explanation]
3. Specificity: [Score] - [Explanation]
4. Practicality: [Score] - [Explanation]
5. Language Appropriateness: [Score] - [Explanation]
Total Score: [1-5 points]

Overall Evaluation:
[Brief summary of the query's strengths and weaknesses]

Improvement Suggestions:
[Provide specific suggestions for improvement if needed]

Conclusion: [If total score ¿= 4, then "Pass"; if total score ¡ 4, then "Needs Improvement"] Please evaluate the following data's querydata['query']

Your response:

---

Figure 13: Query scoring prompt.

## A.10    DATA EXAMPLES

Figure 16 illustrates an example of the Google Translate API. Figure 17 provides an example of a single tool calling task, while Figure 18 demonstrates a repeated multiple tools calling task. Figure 19 shows an example of a parallel multiple tools calling task, and Figure 20 presents an example of a nested multiple tools calling task.

---

**QA Generation Prompt**

You are an expert in using APIs as tools and are highly knowledgeable in multilingual environments. For each task I provide, along with the corresponding API information, you will use the API to complete it.
Your response should follow this format:

```json
{
  "Thought": "Your thought process when facing a problem.",
  "Action": "The API you chose and its parameter information. For example [api_name(parameter_1=\"\",
      parameter_2=\"\"),api_name(parameter_1=\"\",parameter_2=\"\")]"
}
```

Please strictly respond in the language specified by the {country} attribute in the API information.

The API information is as follows:
{api_info}

The task is as follows:
{query}

Your response:

---

Figure 14: Query scoring prompt.

Table 8: Complete distribution of all 29 languages and their task counts.

| Language | Count | Language | Count |
|---|---|---|---|
| en | 12,187 | sl | 24 |
| zh | 3,504 | xh | 22 |
| hi | 513 | ur | 22 |
| fr | 269 | cs | 21 |
| vi | 183 | la | 16 |
| pt | 152 | tl | 11 |
| de | 149 | ms | 9 |
| ja | 119 | it | 9 |
| es | 92 | rw | 8 |
| ar | 89 | fi | 6 |
| id | 82 | hr | 3 |
| th | 61 | gl | 3 |
| ko | 32 | kn | 2 |
| nl | 32 | zu | 2 |
| sw | 29 | | |

---

**Checker Prompt**

You are an advanced AI assistant acting as a strict Quality Checker for a multi-turn, tool using QA dataset. Your primary function is to evaluate the quality of the final assistant's answer in a given dialog.

Your task is to analyze the provided dialog. You must determine if the final "assistant" answer meets our quality standards based on the preceding user question and the tool's response in that turn. Your judgment must be strict. If you find any issue, you must mark it as "Fail".

You will be provided with the following information:
1. **Tool List**: A list of available tools. This provides context on the capabilities the assistant could use.
2. **Dialog**: The complete conversation flow, formatted as a list of turns.
3. **Example List**: A list of pre-judged examples ("Pass" or "Fail"). Use these examples to understand and calibrate your judgment according to our quality standards. Do NOT evaluate the examples themselves.

The final assistant's answer must satisfy ALL of the following criteria to be marked as "Pass".
1. **Consistency**: - **Semantic Alignment**: Does the answer directly and relevantly address the user's latest question? - **Factual Grounding**: Is the answer genuinely and exclusively based on the information from the tool's API response in that turn? It must not contradict the data or omit critical requested information.
2. **Reasonability**: - **No Fabrication**: The answer must NOT contain any information, details, or suggestions that are not present in the API response. It must not hallucinate facts. - **Accurate Reflection**: Does the answer accurately summarize or present the tool's output without exaggeration or misinterpretation?
3. **Linguistic Quality**: - **Fluency & Grammar**: Is the answer fluent, natural-sounding, and grammatically correct? It should be free of awkward phrasing and errors.

Your entire response MUST be a single JSON object with two keys, Here is the Output format:

```json
{
    "decision": "A string, either \"Pass\" or \"Fail\"",
    "reasoning": "A brief string explaining your decision, referencing the specific criterion that
        was violated or confirming that all criteria were met."
}
```

Here are examples to guide your judgment.
{example_list}

Here is a list of tools that will be used in the dialog: {tool_list}

Here is the full dialog: {dialog}

your response:

Figure 15: Checker Prompt.

**API Example**

```
{
  "tool_name": "Google Translate",
  "tool_description": "A tool for translating text between different languages using Google's
       translation services.",
  "home_url": "https://rapidapi.com/nickrapidapi/api/google_translate/",
  "country": "Global",
  "api_list": [
    {
      "name": "translate1",
      "url": "https://google_translate.p.rapidapi.com/",
      "description": "This API endpoint allows users to perform translations of text from one
           language to another.",
      "method": "GET",
      "category": "Utilities",
      "required_parameters": [
        {
          "name": "text",
          "type": "string",
          "description": "The text content to be translated"
        },
        {
          "name": "target_lang",
          "type": "string",
          "description": "The target language code (e.g., 'en' for English, 'zh' for Chinese)"
        }
      ],
      "optional_parameters": [],
      "statuscode": "200",
      "schema": {}
    }
  ]
}
```

Figure 16: Example of Google Translate API.

1242
1243
1244
1245
1246
1247
1248
1249
1250
1251
1252
1253
1254
1255
1256
1257
1258
1259
1260
1261
1262
1263
1264
1265
1266
1267
1268
1269
1270
1271
1272
1273
1274
1275
1276
1277
1278
1279
1280
1281
1282
1283
1284
1285
1286
1287
1288
1289
1290
1291
1292
1293
1294
1295

---

**Single Tool Calling Task Example**

```
{
"system": "You are an advanced AI assistant with expertise in:
1. Intelligent tool and API selection
2. Multilingual problem-solving
3. Systematic analytical thinking
Task Guidelines:
        - Carefully analyze the problem presented
        - Select the most appropriate tool(s) or API(s) to solve the task
        - Provide clear, structured reasoning for your tool selection
        - Ensure precise and efficient problem resolution

Important Principles:
        - If no suitable tool exists, clearly explain the limitations
        - Prioritize accuracy and relevance in tool selection
        - Consider potential alternative approaches
        - Maintain a logical and methodical approach

When tools are insufficient or parameters are incomplete:
        - Explicitly state the gaps in available tools or information
        - Provide constructive suggestions for obtaining resources
        - Return an empty list in the \"Action\" section
Note: Please respond in the same national language as the problem
while maintaining clarity, logic, and structure in your answers.
Below is the list of functions you can invoke (in JSON format):
{api_list}

Please strictly follow the format below, without additional text:

json{\"Thought\": \"Respond in same language as problem. Provide
        analysis, reasoning for tool selection, and challenges.\",
        \"Action\": \"[function_name1(param_1='val1', param_2='val2'),
        function_name2(param_1='val1', param_2='val2')]\"}",
"conversations": [
{
"from": "user",
"value": "Can you help me translate "Hello, how are you?" into French for my
upcoming trip to Paris?"
},
{
"from": "assistant",
"value": "json{\"Thought\":\"To assist the user with their request for
translation, I will utilize the Google Translate API, which is specifically
designed for translating text between languages. The user needs a translation
for a common greeting in French, which is straightforward.\",
\"Action\":\"[Google Translate(text='Hello, how are you?',"target_lang"="fr")]
\"}"}]
}
```

Figure 17: Single tool calling task example.

```
Repeated Multiple Tools Calling Task Example

{
"system": "You are an advanced AI assistant with expertise in:
1. Intelligent tool and API selection
2. Multilingual problem-solving
3. Systematic analytical thinking
Task Guidelines:
        - Carefully analyze the problem presented
        - Select the most appropriate tool(s) or API(s) to solve the task
        - Provide clear, structured reasoning for your tool selection
        - Ensure precise and efficient problem resolution

Important Principles:
        - If no suitable tool exists, clearly explain the limitations
        - Prioritize accuracy and relevance in tool selection
        - Consider potential alternative approaches
        - Maintain a logical and methodical approach

When tools are insufficient or parameters are incomplete:
        - Explicitly state the gaps in available tools or information
        - Provide constructive suggestions for obtaining resources
        - Return an empty list in the \"Action\" section
Note: Please respond in the same national language as the problem
while maintaining clarity, logic, and structure in your answers.
Below is the list of functions you can invoke (in JSON format):
{api_list}

Please strictly follow the format below, without additional text:

json{\"Thought\": \"Respond in same language as problem. Provide
        analysis, reasoning for tool selection, and challenges.\",
        \"Action\": \"[function_name1(param_1='val1', param_2='val2'),
        function_name2(param_1='val1', param_2='val2')]\"}",
"conversations": [
{
"from": "user",
"value": "Can you verify the email address 'john.doe@example.com' using the
Express option? After that, check 'jane.smith@sample.com' with the Premium
option to see if it's valid."
},
{
"from": "assistant",
"value": "json{\"Thought\":\"To verify the email addresses, I will use the
Global Email V4 API. First, I will check 'john.doe@example.com' with the
        Express
option, and then I will verify 'jane.smith@sample.com' using the Premium option
        .",
\"Action\":\"[Global Email V4(opt="VerifyMailbox:Express", email=
"john.doe@example.com"),Global Email V4(opt="VerifyMailbox:ExpressPremium",
email="jane.smith@sample.com")]
\"}"}]
}
```

Figure 18: Repeated multiple tools calling task example.

**Parallel Multiple Tools Calling Task Example**

```
{
"system": "You are an advanced AI assistant with expertise in:
1. Intelligent tool and API selection
2. Multilingual problem-solving
3. Systematic analytical thinking
Task Guidelines:
        - Carefully analyze the problem presented
        - Select the most appropriate tool(s) or API(s) to solve the task
        - Provide clear, structured reasoning for your tool selection
        - Ensure precise and efficient problem resolution

Important Principles:
        - If no suitable tool exists, clearly explain the limitations
        - Prioritize accuracy and relevance in tool selection
        - Consider potential alternative approaches
        - Maintain a logical and methodical approach

When tools are insufficient or parameters are incomplete:
        - Explicitly state the gaps in available tools or information
        - Provide constructive suggestions for obtaining resources
        - Return an empty list in the \"Action\" section
Note: Please respond in the same national language as the problem
while maintaining clarity, logic, and structure in your answers.
Below is the list of functions you can invoke (in JSON format):
{api_list}

Please strictly follow the format below, without additional text:

json{\"Thought\": \"Respond in same language as problem. Provide
        analysis, reasoning for tool selection, and challenges.\",
        \"Action\": \"[function_name1(param_1='val1', param_2='val2'),
        function_name2(param_1='val1', param_2='val2')]\"}",
"conversations": [
{
"from": "user",
"value": "Can you check the latest exchange rates for USD and AUD against EUR,
and also give me a list of all available currencies?"
},
{
"from": "assistant",
"value": "json{\"Thought\":\"To fulfill the request, I need to retrieve the
latest exchange rates for USD and AUD against EUR, as well as a list of all
available currencies. This requires using two different API endpoints: one for
the latest exchange rates and another for the list of currencies.",
\"Action\":\"[latest(from='EUR', to='USD,AUD'), currencies()]
\"}"}]
}
```

Figure 19: Parallel multiple tools calling task example.

1404
1405
1406
1407
1408
1409
1410
1411
1412
1413
1414
1415
1416
1417
1418
1419
1420
1421
1422
1423
1424
1425
1426
1427
1428
1429
1430
1431
1432
1433
1434
1435
1436
1437
1438
1439
1440
1441
1442
1443
1444
1445
1446
1447
1448
1449
1450
1451
1452
1453
1454
1455
1456
1457

**Nested Multiple Tools Calling Task Example**

```
{
"system": "You are an advanced AI assistant with expertise in:
1. Intelligent tool and API selection
2. Multilingual problem-solving
3. Systematic analytical thinking
Task Guidelines:
        - Carefully analyze the problem presented
        - Select the most appropriate tool(s) or API(s) to solve the task
        - Provide clear, structured reasoning for your tool selection
        - Ensure precise and efficient problem resolution

Important Principles:
        - If no suitable tool exists, clearly explain the limitations
        - Prioritize accuracy and relevance in tool selection
        - Consider potential alternative approaches
        - Maintain a logical and methodical approach

When tools are insufficient or parameters are incomplete:
        - Explicitly state the gaps in available tools or information
        - Provide constructive suggestions for obtaining resources
        - Return an empty list in the \"Action\" section
Note: Please respond in the same national language as the problem
while maintaining clarity, logic, and structure in your answers.
Below is the list of functions you can invoke (in JSON format):
{api_list}

Please strictly follow the format below, without additional text:

json{\"Thought\": \"Respond in same language as problem. Provide
        analysis, reasoning for tool selection, and challenges.\",
        \"Action\": \"[function_name1(param_1='val1', param_2='val2'),
        function_name2(param_1='val1', param_2='val2')]\"}",
"conversations": [
{
"from": "user",
"value": "Can you generate a secure password with numbers and then hash that
password using the MD5 algorithm?"
},
{
"from": "assistant",
"value": "json{\"Thought\":\"To generate a secure password with numbers, I will
use the 'Generate password' API with the appropriate parameters. After
generating the password, I will hash it using the 'MD5 Text Hash' API.",
\"Action\":\"[Generate password(numbers=true,pwCount=1,length=12),MD5 Text
Hash(dataString='generated_password')]
\"}"
},
{
"role": "Tool",
"content": {\"code\": 200,\"msg\": \"Password generated successfully.\",
\"data\": {\"password\": \"A1b2C3d4E5f6\"    }}"
},
{
"from": "assistant",
"value": "json{\"Thought\":\"I have generated the password 'A1b2C3d4E5f6'
successfully. Now, I will hash this password using the MD5 algorithm.",
\"Action\":\"[MD5 Text Hash(dataString='A1b2C3d4E5f6')]
\"}"
}
]
}
```

Figure 20: Nested multiple tools calling task example.

