# OpenReview forum: "Enhancing Tool Calling in LLMs with the International Tool Calling Dataset"
_ICLR.cc/2026/Conference — ICLR 2026 Conference Withdrawn Submission_

### Official Review · Reviewer_ef9T · 2025-10-28

**Soundness:** 2
**Presentation:** 2
**Contribution:** 2
**Rating:** 2
**Confidence:** 4

**Summary:**

This work proposes an International Tool Calling (ITC) dataset that comprises of tasks covering 40 countries. The dataset consists of 15790 tasks that are part of the training dataset, and 1750 tasks for evaluation. Based on Table 1, the primary differentiation between the proposed ITC and other tool-calling datasets is that ITC is multi-lingual as opposed to other datasets, which are in the English language.

This work then evaluates multiple open and closed-source models on the benchmark and show that models struggle in selecting appropriate tools or invoking tools appropriately. Fine-tuning models on the training set shows improved performance on the ITC dataset as well as three other benchmarks.

**Strengths:**

The paper makes a meaningful contribution by expanding the toolset beyond English and including multiple languages and geographies. The dataset also includes a sizeable training set, which model builders can leverage to expand their model capabilities to languages beyond English.

**Weaknesses:**

**1. Non-standard function specification in the system prompt:**

Looking at the submitted data, I found that the system prompt contains the function definition (see example below) instead of a separate tools key.  This prevents the tokenizer from applying the appropriate chat template, which in turn negatively affects the model's performance. I suspect that due to this formatting, some of the models have lower-than-expected performance numbers, thus making the evaluation numbers unreliable in my opinion.

Additionally, the function specification is not standard, such as the use of key `api_name, api_description, api_parameters` instead of `name, description, and parameters`. It would be beneficial if the authors could clarify both of these points, and why these formatting decisions were made.

```
You are an advanced AI assistant with expertise in:
1. Intelligent tool and API selection
2. Multilingual problem-solving
3. Systematic analytical thinking

Task Guidelines:
- Carefully analyze the problem presented
- Select the most appropriate tool(s) or API(s) to solve the task
- Provide clear, structured reasoning for your tool selection
- Ensure precise and efficient problem resolution

Important Principles:
- If no suitable tool exists, clearly explain the limitations
- Prioritize accuracy and relevance in tool selection
- Consider potential alternative approaches
- Maintain a logical and methodical approach

When tools are insufficient or parameters are incomplete:
- Explicitly state the gaps in available tools or information
- Provide constructive suggestions for obtaining necessary resources
- Return an empty list in the "Action" section

Note:
Please respond in the same national language as the problem while maintaining clarity, logic, and structure in your answers.

Below is the list of functions you can invoke (provided in JSON format)

[{'api_name': '表情包-百度版API', 'url': 'https://cn.apihz.cn/api/img/apihzbqbbaidu.php', 'api_description': '根据关键词搜索并获取百度表情包', 'required_parameters': [{'name': 'id', 'type': 'string', 'description': '接口盒子分配的唯一ID', 'default': '88888888'}, {'name': 'key', 'type': 'string', 'description': '接口盒子分配的密钥', 'default': '88888888'}, {'name': 'limit', 'type': 'int', 'description': '每页返回的表情包数量', 'default': 10}, {'name': 'page', 'type': 'int', 'description': '当前页码', 'default': 1}, {'name': 'words', 'type': 'string', 'description': '搜索表情的关键词', 'default': '伤心'}], 'optional_parameters': [], 'method': 'GET', 'schema': {}, 'country': 'Global'}, {'api_name': 'Kareem Hesham', 'api_description': 'This API allows users to search for prayer times using a city name or country.', 'required_parameters': [], 'optional_parameters': [{'name': 'Egypt', 'type': 'STRING', 'description': 'Specify the city in Egypt for which to retrieve prayer times.', 'default': 'Assiut'}], 'method': 'GET'}, {'api_name': 'GST Number Search Tool & GSTIN Verification Online', 'url': 'https://gst-details-api-documentation.p.rapidapi.com/GetGSTDetails', 'api_description': 'This API endpoint enables users to search for GST numbers and verify GSTIN (Goods and Services Tax Identification Number) online.', 'required_parameters': [{'name': 'GST', 'type': 'STRING', 'description': 'The GST number to be searched or verified.', 'default': '24AAKCS0993B2ZF'}], 'optional_parameters': [], 'method': 'GET', 'schema': {}, 'country': 'Global'}, {'api_name': 'GET Call', 'url': 'https://nurse-verification.p.rapidapi.com/v3/tasks', 'api_description': 'Fetches the results of the nurse verification process using the request ID received in the initial response.', 'required_parameters': [{'name': 'request_id', 'type': 'STRING', 'description': 'The unique identifier for the verification request.', 'default': '68bbb910-da9b-4d8a-9a1d-4bd878b19846'}], 'optional_parameters': [], 'method': 'GET', 'schema': {}, 'country': 'Global'}, {'api_name': '查汉字字典API', 'url': 'https://cn.apihz.cn/api/zici/chazd.php', 'api_description': '通过汉字查询其相关信息', 'required_parameters': [{'name': 'id', 'type': 'string', 'description': '接口盒子ID，用于标识调用者', 'default': '88888888'}, {'name': 'key', 'type': 'string', 'description': '接口盒子KEY，用于验证调用者身份', 'default': '88888888'}, {'name': 'word', 'type': 'string', 'description': '要查询的汉字', 'default': ''}], 'optional_parameters': [], 'method': 'GET', 'schema': {}, 'country': 'Global'}]Please strictly follow the format below, without any additional text or explanations:
```json{
"Thought": "Respond in the same national language as the problem. Provide a comprehensive analysis of the problem, reasoning behind the tool selection, and potential challenges.",
"Action": "[function_name1(parameter_1='value1', parameter_2='value2'),function_name2(parameter_1='value1', parameter_2='value2')]"
}
```

**2. Lack of executable evaluation metrics**

To create the dataset, the authors validated the APIs by executing them and removing non-responsive APIs or APIs that return an empty response or malformed outputs. Thus, given that the APIs in the dataset are executable, why was an executable metric not included in the evaluation? While the existing metrics of tool selection and invocation are good to include, having an executable metric as the primary metric will improve the dataset.

**3. Deeper analysis on failure modes of models:**

Analyzing models' failure modes beyond what's included in Table 3 will help researchers understand how to improve the models. One of the analyses could be failure rates grouped by language. This particular analysis will tell whether the models fail due to multilingual issues or tool-calling issues.

**Questions:**

See above

---

### Official Review · Reviewer_NN2d · 2025-10-31

**Soundness:** 2
**Presentation:** 2
**Contribution:** 2
**Rating:** 4
**Confidence:** 3

**Summary:**

This paper introduces the International Tool Calling dataset, designed to enhance large language models’ ability to interact with real-world APIs. The dataset contains 3,571 real APIs and 17,540 multilingual tasks across 40 countries and 20 categories, addressing the limitations of previous benchmarks that relied on simulated or English-only APIs. Using models such as GPT-4o, Claude-3.5, and Gemini, the authors built an automated pipeline for API standardization, query generation, and quality filtering. Experimental results show that fine-tuning open-source models on ITC significantly improves tool selection, parameter generation, and cross-lingual generalization—narrowing the gap with top proprietary models like GPT-4o.

**Strengths:**

1. The dataset offers broad multilingual coverage, addressing a major gap since most existing tool-calling datasets are limited to English.
2. It is built on real, publicly accessible APIs rather than simulated or proprietary ones, making the data more authentic, reproducible, and closer to real-world applications.

**Weaknesses:**

Weaknesses:
1. The models evaluated are somewhat dated — including newer ones like GPT-5 or DeepSeek-V3.1 could make the results more representative.

2. The benchmark scores are already very high — for example, Watt-Tool and GPT-4o have achieved excellent results(>80%), which makes it difficult to observe meaningful performance differences across models. In addition, DeepSeek-V3’s Invocation F1 appears even higher than Watt-Tool’s, which might suggest a labeling or reporting inconsistency.

3. The paper provides limited detail on the human verification process in data construction, leading to uncertainty regarding the reliability of automatically generated samples. It is recommended that the authors clarify methodological specifics and provide explicit evidence of quality control and validation results.

4. The fine-tuned models were only tested on the ITC dataset itself; adding evaluations on other out-of-domain benchmarks would make the findings more convincing.

**Questions:**

1.It’s not very clear how the scores for relevance, clarity, and utility were actually used to filter the data — for example, what thresholds were applied, or how disagreements between models were handled. A bit more detail on this filtering process would make the dataset construction more transparent.

---

### Official Review · Reviewer_2YBT · 2025-10-31

**Soundness:** 2
**Presentation:** 2
**Contribution:** 2
**Rating:** 0
**Confidence:** 5

**Summary:**

The paper proposes "LLMs can call tools via APIs, but existing datasets are small, simulated, or lack geographic diversity. ITC offers a real-world, international dataset with 3,571 APIs and 17,540 tasks across 20 categories and 40 countries, built via a four-stage pipeline (API curation, query generation/scoring, and QA creation). Experiments show large gaps between open- and closed-source models, and fine-tuning on ITC boosts generalization, cross-lingual reasoning, and out-of-domain tool use."

The main contributions are: (1) a benchmark of 24 LLMs on ITC that reveals large performance gaps and common tool-use failures (nonexistent tools, missing/incorrect parameters); (2) evidence that fine-tuning on the full multilingual ITC notably improves performance—especially on non-English queries—by enhancing reasoning consistency and cross-lingual generalization; and (3) demonstrated gains in out-of-domain generation and in tool selection/invocation precision on external benchmarks, showing ITC strengthens real-world robustness.

**Strengths:**

The strengths of the paper:

- The dataset is open-sourced and benefits community research in related areas.

**Weaknesses:**

The weaknesses of the paper are listed below,

- Introduction Section:

    - The authors first present ToolLLM as advancing real-world tool invocation, then claim “others use proprietary or inaccessible APIs, as in ToolLLM.” ToolLLM’s core point is using (mostly) publicly documented APIs (often via hubs/marketplaces with keys). Calling it “proprietary/inaccessible” is misleading and internally inconsistent with the prior praise.

    - The “US-centric” generalization + Yahoo-Weather example doesn’t follow. The authors argue “most existing benchmarks are US-centric,” then support it with an API coverage example (Yahoo-Weather not giving district-level data in Shenzhen). That’s a mismatch: API geographic coverage ≠ benchmark design bias. Also, Yahoo Weather is a weak example here (coverage varies and the API itself is not a common tool-use benchmark).

    - Taxonomy inconsistency. The authors say API-BLEND/APIGen/ToolACE “focus on API-based function calling across a variety of use cases,” then say Gorilla/ToolLLM “address real-world tool invocation.” In practice, all of those target real-world API/tool use to some degree; the presented dichotomy isn’t coherent. If the authors intend “closed-loop execution with real endpoints” vs. “static function-calling,” that distinction needs to be made explicit.

    - The authors broadly claim accessibility is hindered because datasets “are not publicly available.” Some well-known tool-use resources are public or partially public (even if keys/quotas are needed). The statement needs narrowing and concrete examples.

- Related Works Section:

    - Inconsistent definitions for “# Tools” and “# Tasks” in Table 1, different works count very different things (API endpoints vs. functions vs. skills vs. trajectories vs. conversations). Numbers like “ToolBench 16,464” and “StableToolBench 16,464” almost certainly refer to instances/trajectories, not distinct tools. Without a unified definition, the column is misleading.

    - In Table 1, at least some items marked × are publicly released (often with keys/quotas). “Publicly available” must be defined (open data files? code + scripts to re-generate? requires API keys?). Paid-key access ≠ “not accessible.”

    - In Table 1, “Primary language = English” is overstated for several rows (many include Chinese/multilingual content). Conversely, the authors' own row uses “Multi-languages” (grammatically odd and inconsistent). Provide explicit language codes (e.g., en/zh-CN) and percentages.

    - In Table 1, there’s no citation or footnote tying each number to a release/version. For volatile corpora, this is essential (and prevents accusations of cherry-picking).

- DATASET CURATION Section:

    - Figure ↔ text stage mismatch. The flowchart labels/query steps don’t match the narrative stages (e.g., Stage 1 says API collection, but the figure’s top-left block is “Query Generation” and there is no explicit “GPT-4o doc supplementation” box).

    - “Minimal human intervention” vs “Checker.” The figure includes a “Checker,” while the text claims minimal human involvement. If the checker is human, say so and quantify; if it’s an LLM, call it that.

    - Evaluation leakage risk. GPT-4o (and Claude/Gemini) are used to author API instructions, refine queries, and generate QA pairs, and later are used for testing. That creates obvious information leakage / contamination and undermines fairness unless strict model isolation is enforced.

    - “Real API” claim vs. LLM-rewritten specs. The authors say the corpus contains real APIs but also that GPT-4o “completes/refines incomplete specs.” That means parts of the specification are synthetic; it weakens the “real-world spec” claim unless these portions are marked and excluded from evaluation.

    - Unclear feasibility of large-scale execution. Testing ~49,937 APIs with scripted calls (3–5 calls each × two stages, over 48 hours) is logistically dubious—most endpoints require authentication/keys, quotas, captchas, or payments. The text does not explain how keys were provisioned or how auth-gated APIs were handled (biasing toward no-auth endpoints).

    - “Update frequency (inactive >12 months)” is poorly defined. Many stable APIs don’t publish update timestamps; marketplaces’ “last updated” fields are not standardized. It’s unclear how the authors measured this and whether it’s a valid quality criterion for endpoint usability.

    - Stability inference from 3–5 calls/48h is weak. That sampling can’t establish reliability; it also can’t capture diurnal/weekly variability or rare failure modes. Calling this “stability” is overclaiming.

    - Source list is shaky/ambiguous. Names like “Free-api,” “Public-apis,” “XiaRou,” etc., are not standard identifiers; some are lists, not providers. The authors need precise, citable sources and versions. Otherwise the “49,937 APIs” figure lacks provenance and is likely inflated or double-counted.

    - Executability vs. documentation quality conflation. The authors verify “correctness through sample executions” yet also rely on LLM-completed specs; a spec can be “consistent” but wrong while the sample call coincidentally works. The authors need stronger validation.

    - Claim of “enhanced usability for developers.” If specs are partly LLM-generated, that claim is unsupported unless the authors ran human-dev usability studies.

    - Unclear separation between training data and eval. The authors say the pipeline “generates high-quality QA pairs” (used later for testing). If the same LLM family authored both instructions and QA, eval becomes self-generated rather than independent ground truth.

    - Coverage thinness / unclear multilinguality. “Three user queries per API (or set)” is unlikely to cover API functionality or edge cases, and “multilingual repository” is not defined (which languages? how ensured per-API support?).

    - Overclaims about human review. “Exhaustive human review” of 18,368 queries with five annotators (each query double-checked on five criteria) is a very large workload, yet no inter-annotator agreement (κ/α), time per item, or quality control is reported. That weakens the reliability claim.

    - Arbitrary thresholds w/o justification. Keeping only items “scoring >4 from both models” (1–5 scale) is an extremely aggressive precision-oriented cut that likely destroys recall and language diversity; no analysis shows why this is appropriate.

    - Human–LLM role ambiguity. “Checker” is used again in section 3.4 but not defined (human or LLM?). You also say “low-complexity issues were corrected directly by humans,” but the bulk acceptance rate (only 6.9% modified) after heavy LLM involvement suggests circular validation rather than independent adjudication.

- Experiments Section:

    - Claude, Gemini, and GPT-4o were used during dataset construction and again during testing, which may introduce information leakage and unfair comparisons, thereby biasing the experimental conclusions (as I mentioned above).

    - Judging from Figure 7, the paper’s touted multilingual support is extremely imbalanced—many regions have only single-digit tools—so it would be more practical to just add a translation layer.

**Questions:**

The specific issues are listed in the Weaknesses section. Overall, the paper reads as mediocre: many design choices in dataset construction lack experimental validation. The multilingual motivation appears overstated; based on the data distribution, performance in many languages may be worse than simply adding a translation layer, which would also cover more APIs across languages. Moreover, there are signs of information leakage between training and testing, making the results unreliable. Taken together, these concerns lead me to recommend rejection.

---

### Official Review · Reviewer_6zzm · 2025-11-01

**Soundness:** 2
**Presentation:** 3
**Contribution:** 2
**Rating:** 4
**Confidence:** 3

**Summary:**

This paper introduces the International Tool Calling (ITC) dataset, a large-scale, multilingual, and geographically diverse benchmark for evaluating and improving tool-calling capabilities in large language models (LLMs). ITC comprises 3.5k real-world APIs and 17.5k tool-calling tasks across 20 categories and 40 countries. The dataset is constructed through a semi-automated pipeline involving API collection, query generation, scoring/filtering, and QA pair creation, with both LLMs and human annotators involved in quality control. The authors benchmark many open-source and closed-source LLMs, showing significant performance gaps and highlighting the benefits of fine-tuning on ITC for cross-lingual and out-of-domain generalization.

**Strengths:**

1) The proposed ITC dataset is valuable for the community, considering its size, multilingual coverage (29 languages), and inclusion of both global and region-specific real publicly accessible APIs.
2) The paper benchmarks a wide range of models using multiple metrics (tool selection, invocation, format, and language matching), and provides error analysis and ablation studies.

**Weaknesses:**

1) The benchmark results show GPT-4o achieving the best performance, while o1-mini and o3-mini perform worse (even below GPT-4o-mini). This raises concerns about the reliability and fairness of the evaluation, especially since the answers in the dataset are generated by GPT-4o, potentially introducing bias in favor of this model. The paper does not systematically analyze or discuss this bias, nor does it provide sufficient ablation or sanity checks (e.g., cross-model answer generation, human evaluation) to quantify its impact. The counterintuitive quality rank make me doubt the data quality.

2) The reliance on free, public APIs means that the dataset is inherently unstable—API endpoints, parameters, and outputs may change or disappear over time. This undermines the reproducibility of the benchmark and the long-term value of the dataset, especially since the main contribution is the dataset itself. The paper acknowledges this limitation but does not propose concrete solutions or mitigation strategies.

3) While the error analysis is detailed, the discussion of why certain models (especially o1-mini/o3-mini) underperform is limited. There is little insight into whether the issues are due to model architecture, training data, or evaluation bias.

4) Finetuning on the training split can easily boost the model performance to be very high, which reflects the possible annotation bias and the real difficulty level.

**Questions:**

Refer to the above part.

---

### Note · Authors · 2026-01-09

I have read and agree with the venue's withdrawal policy on behalf of myself and my co-authors.